# The Impact of Training on Druze Entrepreneurs' Attitudes Towards and Intended Behaviors Regarding Local Sustainability Governance: A Field Experiment at the Mount Carmel Biosphere Reserve

**Itai Beeri [1,\*]**, **Dan Gottlieb [1,2]**, **Ido Izhaki [3]**, **Tzipi Eshet [4]** and **Noam Cohen [1]**

1    School of Political Sciences, University of Haifa, Haifa 3498838, Israel; dang@envihaifa.org.il (D.G.); nc.noamcohen@gmail.com (N.C.)
2    Haifa Bay Municipal Association for Environmental Protection 31250, Haifa 3498838, Israel
3    Department of Evolutionary and Environmental Biology, University of Haifa, Haifa 3498838, Israel; izhaki@research.haifa.ac.il
4    Natural Resources and Environmental Research Center, University of Haifa, Haifa 3498838, Israel; tzipi.eshet@gmail.com
\*    Correspondence: itaibeeri@poli.haifa.ac.il

**Abstract:** This article expands our understanding of biosphere reserve management by exploring the effect of green business-guidance training. Biosphere reserves promote conservation while enabling sustainable use by local communities, in keeping with the notion of local sustainability governance. In practice, however, many local communities regard biosphere reserves as an obstacle to their economic growth and prosperity, resulting in active resistance to them. Given this complexity, we ask whether green business-guidance training changes entrepreneurs' attitudes towards and intended behaviors regarding local sustainability governance. To test this question empirically, we used action research and designed a before-and-after field experiment in the Mount Carmel Biosphere Reserve in Israel. Our findings suggest that green business-guidance training has a significant positive effect on entrepreneurs' attitudes towards and intended behaviors regarding local sustainability governance and that biosphere reserve managements can improve local sustainability governance performance via training. We discuss our findings and suggest new paths for research in theory and practice.

**Keywords:** biosphere reserves; conservation management; environmental entrepreneurship; local sustainability governance; field experiment; organizational behavior; local communities; minorities

## 1. Introduction

According to the United Nations Educational, Scientific and Cultural Organization [1], biosphere reserves are areas comprising terrestrial, marine and coastal ecosystems that seek to promote conservation while enabling sustainable use by local communities. UNESCO defines biosphere reserves as "'Science for Sustainability support sites'—special places for testing interdisciplinary approaches to understanding and managing changes and interactions between social and ecological systems, including conflict prevention and management of biodiversity." According to UNESCO, as of 2019 there were 701 biosphere reserves in 124 countries around the world, including 21 *trans*-boundary sites. Biosphere reserves are designated by national governments and managed by a varying array of systems, with the aim of reconciling conservation goals with the economic and social needs of local communities. In practice, however, many local communities, especially indigenous minorities, regard biosphere reserves as an obstacle to their economic welfare and a threat to their future development,

resulting in a lack of cooperation and even active resistance to their existence, goals and activities [2,3]. Thus, one of the main challenges for biosphere reserves is to integrate local communities into the management and development of the reserve, fostering their sense of belonging while meeting their social and economic needs and aspirations [4,5].

Israel has two designated biosphere reserves: the Mount Carmel Biosphere Reserve (designated in 1996) in the northwest of Israel, and the Ramot Menashe Biosphere Reserve (designated in 2011), located southeast of Mount Carmel. Since its designation, the Mount Carmel Biosphere Reserve has been managed by national environmental regulators without the participation in decision-making of its largest and most dominant population, the Druze community, which constitutes more than 29,000 of the 60,000 residents in the Carmel Biosphere Reserve. In fact, research in 2014 found that only eleven percent of all residents living in the Mount Carmel reserve were even aware of the area's designation [6]. The result has been friction, mutual mistrust, resistance and hostility, which endanger the reserve's future status [7].

In 2010, the University of Haifa established the Center for Carmel Research with the aim of promoting and strengthening the Mount Carmel Biosphere Reserve through interdisciplinary research and the active engagement of local stakeholders, including residents, local authorities and other institutions [8]. Preliminary discussions confirmed that the Druze community was not included in the decision-making related to the Mount Carmel Biosphere Reserve, and that attitudes among the Druze towards the reserve and its management were strongly negative. However, these preliminary inquiries also suggested that involving the Druze community in the sustainable development of the area and in local economic entrepreneurship might improve their attitudes and reduce the community's active resistance to the reserve. Accordingly, we conducted an action research study aimed at exploring whether and how a training program for local entrepreneurs might encourage positive attitudes towards and intended behaviors regarding local sustainability governance within the Druze community concerning the Mount Carmel Biosphere Reserve. More precisely, this study sought to conceptualize and empirically test the relationship between attitudes towards and intended behaviors regarding sustainability governance related to the Mount Carmel Biosphere Reserve and the effect of green business-guidance training on these relationships among the Druze and Druze entrepreneurs.

While biosphere reserves have been in existence for many decades, the focus on fostering sustainable social and economic development dates only to the late 1990s [9–11]. Nevertheless, these ideas have appeared in various theoretical and practical contexts, such as the applied ecological sciences, social sciences and eco-philosophical works [12]. Since then, there has been no systematic, empirical investigation of whether and how a conservation program or training program might affect the attitudes towards and intended behaviors of entrepreneurs regarding local sustainability governance and the concept of the biosphere reserve. The current study thus answers the call from several scholars (e.g., [12–15]) to explore the relationship between sustainability governance, management and the local community. Thus, we selected the systematic review approach to present an overview of the field of sustainability research that provides an in-depth account of conservation programs and their relationship with local sustainability governance and the concept of the biosphere reserve.

Furthermore, the present study has a unique contribution to theoretical and applied research of local sustainability development. By definition, sustainability is a concept that combines environment, economy and community base on the assumption that environmental welfare goes hand in hand with social and economic well-being. Still, many studies in the field of sustainability, seek to examine lifestyle changes as a key to achieving sustainability in the environmental aspect. This is by adopting a low carbon lifestyle based on reducing the consumption of material and energy in order to benefit the natural environment. The present study seeks to foster local sustainability through entrepreneurship and strengthening a local economy. The assumption is that, if residents of the biosphere reserve recognize the economic potential of the reserve, the result will be the strengthening of the local economy, and the expression of positive attitudes of the residents towards the biosphere reserve. In other words,



the realization of the economic potential inherent in the biosphere reserve can therefore make a significant contribution to strengthening local sustainability in both environmental and community aspects.

## 2. Theoretical Background

### 2.1. Land Regulation and Spatial Inequality

Under the prevailing approach to public policy, the regulation of space and economic development for local authorities and communities requires state authorities to design and determine the boundaries between natural and environmental assets [16]. State regulation of economic, social and environmental interests by setting boundaries and managing space influences the proliferation of businesses and the socioeconomic growth of communities [12,17]. The regulation of land resources should balance unrestrained competition between local authorities and communities, and thereby support inclusive growth, while reducing spatial inequality.

In Israel, ethnic tensions are strongly rooted in the centralized system of land control and land rights, which, in turn, have created structural spatial inequality. Since the country's foundation, the map of local government has given preference to local authorities that are populated by Jews and located in the country's center, while tending to neglect local authorities located in the geographic periphery and populated by those on the margins of society—non-Jews, minorities, immigrants, and residents of low to medium socio-economic status [18]. The lack of local control over land rights has combined with sharp increases in population growth and density, and the fact that half of the country's land is resource-poor, to fuel economic and spatial inequalities [19,20]. Considering these challenges, biosphere reserves have the potential to reduce spatial inequality if they are managed appropriately, or to worsen it if they are not.

### 2.2. From Conservation to Local Sustainability Governance: The Concept of the Biosphere Reserve

The creation and promotion of environmental conservation are not new phenomena. For instance, national parks in Canada and the United States, such as Banff and Yellowstone, were established in the nineteenth century. Their creation reflected a "protection" ethos that saw nature as wild and pure, in contradistinction to destructive human development and industrialization [9,21]. This approach focuses on the natural world and its "internal values" separately from human beings and their needs. Its practical expression is the establishment of nature reserves intended primarily to preserve plants, animals and landscape configurations in their existing, undeveloped state. In this model, the very idea of development is invalid [22]. As such, while, in many areas, the concept of the nature reserve enabled the rescue of local biodiversity and natural resources with unique value, elsewhere the nature reserve is regarded as restricting local communities from developing by denying them access to local natural resources (e.g., [23]).

In recent decades, perceptions of environmental preservation in Western society have changed to include the idea that human beings and their needs should be considered an integral part of the biological diversity equation. Unlike the classic biodiversity conservation approach, under which conservation was considered to be opposed to development, the biosphere reserve approach recognizes the importance of taking the economic development of the community and cultural values into account [2,3,9,24]. As Hawkins, Kwon and Bae [25] noted, while, at the national level, the principles of sustainability mean using economic instruments, such as taxes or subsidies to incentivize environmentally sound behavior by residents and businesses, at the local level, environmental conservation initiatives are used to promote sustainability alongside economic development. Hence, biosphere reserves are sites for sustainable development with conservation integrated into their management. They are established by countries and recognized under UNESCO's Man and the Biosphere program to promote sustainable development based on local community efforts and sound science [15,24,26].

Since the formulation of the biosphere reserve concept in 1971, there has been ongoing debate over the appropriate roles for existing communities in safeguarding and enhancing biodiversity

conservation [9]. An important change took place in 1996 with the adoption of the so-called Seville Strategy and a new statutory framework for biosphere reserves, which expanded the reserves' remit to include fostering sustainable social and economic development [9–11]. The biosphere reserve concept has evolved to represent an interdisciplinary philosophy that addresses the ecological, social, and economic dimensions of biodiversity loss, which we call sustainability governance. Following Meadowcroft, Farrell and Spangenberg [27], we define sustainability governance as governance oriented towards sustainable development. It includes regulatory mechanisms and governmental practices designed to promote social development along with sustainable development. We maintain that sustainability governance can be promoted by biosphere management and training in how to do so. Such training emphasizes the relationship between local stakeholders, such as communities, individuals, business and NGOs, and biodiversity and the conservation of socio-ecological values alongside natural systems [28]. Thus, managing a biosphere reserve requires conservation authorities to reconcile the goals of conservation and economic development in the same space, and to foster all of the stakeholders' long-term interests [29].

In accordance with this concept, all biosphere reserves are designated by the national government and are expected to fulfill three functions: (a) conserve biodiversity; (b) foster sustainable social and economic development; and (c) support research, monitoring and education. In practice, biosphere reserves follow an innovative land management strategy based on a hierarchy of conservation levels from the center to the periphery: a core zone, a buffer zone and a transition zone. The core zone is where environmental conservation is maximized. The function of this area is solely to preserve the local biodiversity that characterizes the unique ecosystem of the reserve. The middle zone, or buffer zone, protects the core from development pressure. Only activities such as research and monitoring —meaning, "soft" development—are permitted in this area. Finally, the outer zone, or transition zone, allows for the rational use of local natural resources and environmental activities for the economic benefit of local communities, with the overall goals of the biosphere reserve in mind. For instance, in this area, conservation authorities that are in charge of biosphere management and sustainability governance may hire local residents for maintenance and rehabilitation jobs based on natural and heritage resources, with the aim of instilling environmental consciousness in local communities. In a similar vein, conservation authorities may invite and encourage local businesspeople and entrepreneurs to invest in eco-tourism initiatives and promote relatively small-size businesses that cohere with the concept of sustainability [2].

In this sense, the biosphere reserve concept has the potential to realize the goals of sustainability governance, such as strengthening local communities and initiatives, developing local economies and preserving natural assets, without prioritizing environmental conservation at the expense of socio-economic development [30]. Hence, the day-by-day tasks of biosphere reserves are typically cross-sectoral and link the responsibilities of international, national, regional and local institutions (e.g., agriculture and forestry authorities, water management authorities, coastal protection and land use authorities and national park authorities and local governments). As a result of these inter-sectoral, inter-organizational and inter-disciplinary activities, coordination, cooperation and collaboration can be considered one of the main tasks and challenges of local sustainability governance and biosphere reserve management [23,31,32].

Some scholars have critiqued the participation paradigm, asserting that it erodes some aspects of democracy and increases the tension between professionalism and representation. For instance, Lawhon and Patel [32] question the continued viability of addressing sustainability concerns at the relatively small local scale. Bäckstrand [33] highlights debates about the costs and impact of sustainability governance that might increase democratic deficits by weakening the legitimacy of formal regulatory bodies in the eyes of the public, and by questioning the regulators' accountability due to the greater scrutiny and inspection that usually take place in participation processes. Brody [34] discusses the risk that participation can lead to conflicting interests, slow down decision-making and result in unwanted compromises between pure biodiversity conservation and popular economic development.

Other scholars have argued that in a human-dominated world, the goals of biodiversity conservation and economic development are inherently in competition, and that exaggerated representations of economic interests have negative consequences for biodiversity [35].

Contrary to widespread expectations, many biosphere reserves have failed to promote local participation successfully and put the open-reserve concept into practice. The results are misunderstandings, lack of support and conflicts with inhabitants and local stakeholders [3,15]. Accordingly, García-Frapolli et al. [36] suggested the impairment model for conservation conflicts, which maintains that the existence of disagreements between people and/or institutions does not automatically translate into a conflict. A conflict will exist only if one of the parties feels "impeded or diminished" by the behavior of the other party. This approach helps frame the real conflict, map it, manage it and generate solutions. Doyon and Sabinot [2] maintain that biosphere reserves have not succeeded in instilling a substantial environmental and conservation ethos in local residents for two reasons: first, because some members of the local community fail to make a sufficient living from the reserve and so seek alternative sources of income; and second, because members of the local community may be asked to participate in the reserve's activities, but are not invited to take part in the relevant decision-making processes. Thus, it remains a challenge for biosphere reserve managements to consider the social, economic and cultural needs of the local community.

### 2.3. The Case of the Mount Carmel Biosphere Reserve—from Ecological Rehabilitation to Crisis Management

The great fire that occurred in the Carmel mountain range in 1989 was a catalyst for the establishment of the first biosphere reserve in Israel. The fire destroyed 530 hectares of woodland and forest and led to the establishment of a special committee to rehabilitate the damaged area.

The Carmel range was declared a biosphere reserve by UNESCO in 1996. The reserve has a total surface area of 26,600 hectares (see Figure 1). The Mount Carmel Biosphere Reserve, like other biosphere reserves around the world, has value from an ecological, tourist, economic and community perspective. It is one of the largest open spaces in northern Israel, and is a typical example of a Mediterranean ecosystem that includes a rich inventory of geological phenomena, prehistoric artifacts, biodiversity and landscapes [37].

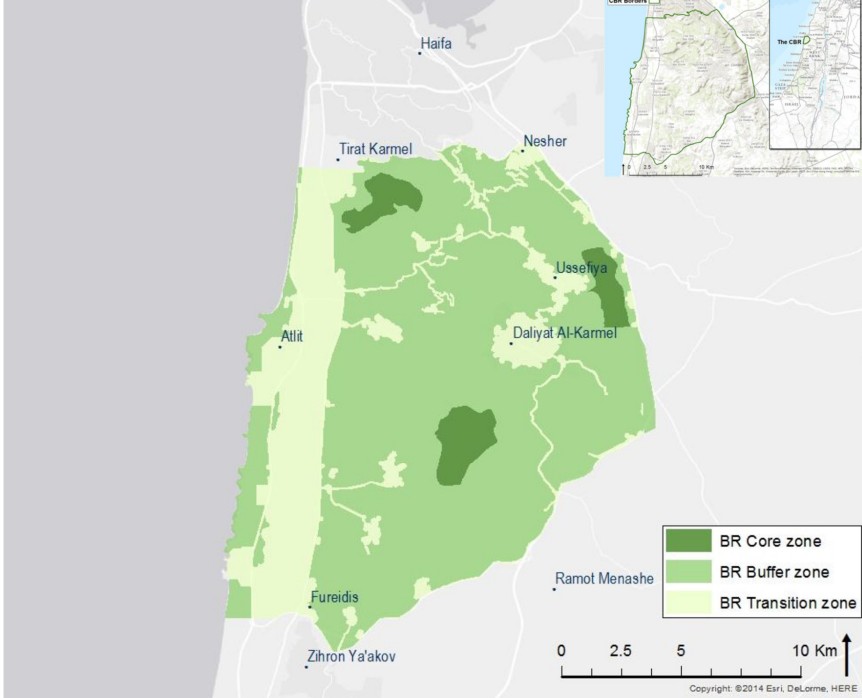

**Figure 1.** Mount Carmel Biosphere Reserve. Note: clockwise, from top right: [38].

From an anthropogenic perspective, the Mount Carmel Biosphere Reserve is unique and diverse. The total population numbers about 60,000, including residents of urban municipalities alongside rural villagers, religious people alongside secular and Jews alongside Muslim or Christian Arabs and Druze. The Druze comprise the majority of the population (about 29,000, or 59%) and live in the biggest towns in the reserve, Daliyat el-Carmel and Isfiya, which are also the two largest (of 18) Druze towns in Israel.

For centuries, the Druze minority maintained its traditional character and a form of rural settlement, including areas of agriculture and grazing, which served as the community's main source of livelihood. However, free-market nations are unlikely to develop rural village settings, and poorer members of centralized industrial societies are more likely to pay the costs of both deforesting and conservation programs [12]. Accordingly, in recent decades, against the background of both economic growth and population growth in the state of Israel, there has been growing protest within Druze communities against the expropriation of their privately owned lands by the state for the construction of infrastructure projects and nature conservation. In addition, the growth of the Druze population has created a demand for land for construction in Daliyat el-Carmel and Isfiya.

However, these needs have not been met. Broadly speaking, sustainability governance in the Mount Carmel Biosphere Reserve has not succeeded in implementing the regulatory mechanisms and practices that promote the social development of local communities, individuals and businesses, along with sustainable development. From its inception, the Mount Carmel Biosphere Reserve was managed by a joint administration consisting of a number of national regulators and governmental environmental conservation organizations—the Ministry of Environmental Protection, the National Parks Authority, the Nature Reserves Authority and the Jewish National Fund (In 1998 the National Parks Authority and Nature Reserves Authority merged to form the Nature and Parks Authority. The Nature and Parks Authority ended its management role with the Mount Carmel Biosphere Reserve in 2019). Druze residents, municipal representatives and other local representatives of commercial and communal civil society groups were never included in the body set up to manage the biosphere reserve, nor have the Druze community and local authorities been active partners in planning, designing and conserving the reserve. Thus, the Mount Carmel Biosphere Reserve management failed to promote a common, integrated vision that rose above organizational interests or individual stakeholder concerns. In practice, the management system was not a catalyst for collaboration, did not function well and rarely expressed its ideas of governance. As a result, many Druze residents oppose the Mount Carmel Biosphere Reserve, which they regard as providing a reason for the government to expropriate private lands [8].

## 3. Research Model and Hypotheses

### 3.1. Rationale for the Study

As discussed above, in the mid-1990s UNESCO reframed the goals and purpose of biosphere reserves to include fostering sustainability governance and supporting research, monitoring, and education, along with conserving biodiversity and protecting the natural environment [9–11]. In accordance with this change, the focus of research in the field of biosphere reserve management shifted and expanded. Whereas, initially, the biosphere reserve program researchers tended to focus on flora, fauna and ecological challenges, over the last two decades, a new wave of research has explored management challenges from an anthropogenic point of view, including combining the social sciences and the ecosystem service concept (e.g., [12,15]). This group of studies examines the role of human dimensions and ecosystems in biosphere management in the realm of sustainability governance.

Based on this approach, we developed our study using several assumptions. First, we assume that sustainability governance—as a concept, ideal and practical regulatory tool—exists to some extent in any context. Second, we maintain that within local communities living in biosphere reserves, the most practical mechanism of sustainability governance is the biosphere reserve management. Third, we assume that the perceptions and attitudes of the local communities living in biosphere

reserves towards the concept and values of the sustainability governance, both positive and negative, will be affected by the biosphere reserve management's policies about and actions regarding social development along with sustainable development. In addition, this effect is more likely to be positive if the management involves the public in its decisions. Fourth, we maintain that the perceptions and attitudes of local communities living in biosphere reserves towards sustainability governance, both positive and negative, are related to the intended behaviors of those responsible for the governance. Fifth, local entrepreneurs are an essential part of the community, and may have a greater social impact than their individual actions. They are highly motivated and skilled at achieving societal and commercial goals. They are individuals who exploit opportunities to influence their surroundings and to increase their self-interests using innovative ideas and strategies [39]. Thus, interventions, programs or training provided to local entrepreneurs by the biosphere reserve's management, and that includes meetings and sessions with its decision makers and policy makers, will increase sustainable businesses in the transition zone of the biosphere reserve. In turn, the integration of the reserve's economic entrepreneurship community will strengthen positive attitudes towards the reserve and its managers and healthy biosphere reserves in general. Improvements in these areas should increase local participation in decision making, improve conservation efforts and lead to balanced development. Thus, we expect training to increase local entrepreneurs' positive attitudes towards and intended behaviors regarding sustainability governance. We also maintain that those best positioned to provide such training are those who work for the biosphere reserve management. Based on this rationale, we developed two hypotheses.

### 3.2. The Relationship between Residents' and Entrepreneurs' Attitudes Towards and Intended Behavior Regarding Local Sustainability Governance

The well-known theory of planned behavior stresses that the strength of attitudes towards a behavior, alongside intentions and subjective norms, serves as an immediate antecedent of behaviors and helps account for considerable variance in actual behaviors [40]. This theory has been widely used in various contexts, including the entrepreneurship literature [41]. People's attitudes towards a behavior—beliefs that a certain behavior will lead to a favorable or unfavorable outcome—help determine the intensity (i.e., the time and effort) with which they engage in that behavior [40,42].

Using the planned behavior theory, we can expect a direct relationship between the attitudes towards and intended behaviors of Druze and Druze entrepreneurs regarding sustainability governance related to the Mount Carmel Biosphere Reserve. The designation of an area as a biosphere reserve does not just carry international recognition, but also entails a long-term commitment to social, economic and ecological responsibility. Hence, it is quite likely that conservation programs will affect the beliefs and actions of local communities that live within the reserve's boundaries [28,29], particularly the rural poor [12].

In addition, pragmatically speaking, people are more supportive of conservation if they believe they will benefit from it [14,15,43]. The opposite is also true. People will be less supportive of conservation if they believe that living close to conservation sites involves a loss of their rights and resources. Local communities living in biosphere reserves, especially relatively small groups who interact over long periods and through reciprocal relations [12], often face a choice between collaboration and creating real alliances on the ground, or rejecting the idea of collaboration. In the latter case, they face losing access to natural resources, and their activities aimed at regaining their sources of subsistence become criminalized.

Applying this argument to the present case, members of the Druze community, and Druze entrepreneurs in particular, are more likely to balance economic development with societal and cultural values, and to accept the concept of a biosphere reserve and sustainability governance based on ecological, social and economic principles, if they regard these values and concepts as important and positive. Only then will they act to achieve the local sustainability governance goals of the biosphere reserve—strengthening the local community, developing the local economy and preserving natural

assets—by coordinating, cooperating and collaborating with stakeholders, such as the biosphere reserve's management [3,26,28–31]. In a similar vein, members of the Druze community, and Druze entrepreneurs in particular, are more likely to actively oppose and obstruct these sustainability goals if they have negative attitudes towards the biosphere reserve and its management. In other words, they will oppose the project if they regard environmental conservation as coming at the expense of their socio-economic development, feel they have no voice in managing the reserve, and feel that the reserve threatens their culture and traditions [8,21,22,30]. We therefore hypothesize that:

**Hypothesis (H1).** *There will be a direct relationship between attitudes towards local sustainability governance and intended behaviors regarding sustainability governance.*

*3.3. The Effect of Green Business-Guidance Training on Entrepreneurs' Attitudes and Intended Behaviors Regarding Local Sustainability Governance*

The governance approach joins a stream of work challenging the classic top-down approach that assumes a clear dichotomy between politics, administration and policy making [44–46]. In accordance with the realm of governance, promoting informed policies is vital for improving the effectiveness of policy outcomes [47]. Thus, the literature suggests that practical guidance for stakeholders in the public sphere increases the likelihood of improved behavioral change [48].

In a similar vein, the sustainability governance concept underscores the interdependence between ecosystems and society. It implies that place-based policies, an ecosystems orientation, collaborations and networks are likely to have a greater impact on green policy implementation than strict protectionism based on authoritarian practices [15]. However, the effectiveness of collaborative sustainability governance depends, inter alia, on administrative capacity and stakeholders' support. Thus, building dialogue and encouraging participation among and with local stakeholders appears to be one of the preliminary conditions needed to set up biosphere reserves and manage them from a sustainable development perspective [33]. As Heinen [12] put it, the question is how to convince people to promote sustainability if they do not regard it as being in their own short-term interests.

A line of research has recently emerged in the business and public administration literature on the effect of entrepreneurship training (e.g., [49,50]). Within the domain of entrepreneurship, guidance may provide knowledge about actors, policy problems and policy solutions related to sustainability governance. Local entrepreneurs are motivated to and skilled at accomplishing societal and commercial goals. They know how to exploit opportunities and use innovative ideas and strategies to influence their surroundings and improve their self-interests [39]. Accordingly, we expect that if the conservation authorities provide training to local entrepreneurs, including meetings and sessions with the authorities' decision makers and policy makers, they will improve the local entrepreneurs' attitudes towards and intended behaviors regarding sustainability governance. Furthermore, we claim that, since a change in moral belief alone is insufficient for solving environmental problems and changing behaviors, individuals and communities should be provided with other rewards and alternatives [12]. One example is green business training guidance, designed to provide rewards for the local community. As studies have demonstrated, the greater the rewards, the more likely the success in changing attitudes towards and intended behavior regarding sustainability [13]. Thus, training may provide practical tools by forming coalitions, working with teams, networking with decision-makers and interacting effectively with policymakers, and promoting local businesses or initiatives. Practically speaking, training can affect the attitudes and intended behaviors that are related to sustainability governance [51,52]. Building on this rationale, we hypothesize that:

**Hypothesis (H2).** *Green business-guidance training will produce positive attitudes towards and intended behaviors regarding local sustainability governance.*

## 4. Method

### 4.1. Procedure

To test these hypotheses, we used a longitudinal design, with data collected in two waves—before and after the green business-guidance training. At T1 (early 2014) a survey was randomly distributed among 265 Druze residents of two local councils, Isfiya and Daliyat el-Carmel, located in the Mount Carmel Biosphere Reserve in northwestern Israel. To recruit participants, research assistants approached passersby in public and commercial areas at various hours during the day. Following the initial data collection, 35 Druze entrepreneurs were selected using the snowball method (an original set of informants were asked by the researchers to list others who met the criteria, and then these candidates were interviewed and asked to list additional others) to participate in a unique green business-guidance training (hereafter, the training), especially designed and adapted for the characteristics of the Mount Carmel Biosphere Reserve (see below). These 35 participants completed the same survey as the general population of participants before the training (T1). All 35 participants completed the training program. After the training was completed, the survey was distributed again to the group of entrepreneurs (T2, mid-2015). As a result of difficulties reaching all 35 participants, only 20 entrepreneurs returned completed T2 surveys.

Based on our hypotheses, we proposed that at T1, before the green business-guidance training, there would be no significant differences between the general Druze population and the Druze entrepreneurs in their attitudes towards and intended behaviors regarding local sustainability governance. At T2, we expected that in comparison with the general Druze population, Druze entrepreneurs who participated in the training would have significantly more positive attitudes towards and intended behaviors regarding local sustainability governance.

### 4.2. Action Research and the Green Business-Guidance Training Program

Action research is a research tool that aims to effect changes in culture, attitudes and behaviors in organizations and communities. It requires learning, practice and new behavior. Action research is unique among research methods, in that the researcher is directly involved with the participants. The dialogue with participants may strengthen and empower the individual, the organization and the community, and encourage the implementation of targeted change [53,54].

In accordance with the principles of action research, the steering committee of the training program consisted of the investigators; the Chief Scientist at the Ministry of Environmental Protection; a representative of the Nature and Parks Authority; a representative of the Keren Kayemeth LeIsrael–Jewish National Fund (KKL–JNF), which is dedicated to developing the land of Israel; the Isfiya Local Authority; the Daliyat el-Carmel Local Authority; and two NGOs—the MATI Haifa Business Development Center and the Carmelim Tourism Association. Thus, the steering committee itself, the professional trainers who operated on its behalf, the training sessions and the professional meetings with the conservation authorities' decision makers and policy makers constituted the regulatory and practical mechanism of sustainability governance. In other words, this group was a government tool that promoted social development along with sustainable development in the name of the biosphere management.

The training was aimed at guiding and empowering local Druze entrepreneurs who were seeking to develop and promote local small and medium-size enterprises that were in keeping with the goals of the Mount Carmel Biosphere Reserve and compatible with the spirit of sustainability governance. For example, businesses might involve local tourism (e.g., guest houses, restaurants, physical sport activities); agriculture and pastoralism (e.g., grazing, felling, wastewater reuse, ecological agriculture); or history and heritage (e.g., the restoration and preservation of historical sites, or leisure activities aimed at instilling the Druze heritage).

The training was implemented on behalf of the steering committee by the management and professional expert team of the Carmelim Tourism Association. All operations and sessions led by the trainers promoted the values and principles of the biosphere reserve and sustainability governance with the aim of encouraging positive attitudes and behaviors that were in line with these ideas. The training comprised two multidimensional phases. Each phase included a subphase of learning and planning, and a subphase in which participants implemented what they had learned, either individually or collectively. The first phase (through 2013) focused on diagnosing the potential opportunities and barriers to the development of local businesses that matched the values of the biosphere reserve. This learning and development process took place in collaboration with representatives of the Druze population (mainly local authority managers in charge of environmental conservation), regulators from the government bodies and the researchers, and allowed for the development of a training program specifically tailored for the Mount Carmel Biosphere Reserve. In the second phase (early 2014 to mid-2015), participants were given 100 h of tailored training, including (a) personal counseling and coaching in financial management, organizational management, building a business plan and project management; and (b) group counseling and coaching in formulating a product, marketing in the new media era, economic planning and dealing with policy makers and local regulators. Both phases of the training took place in Isfiya and Daliyat el-Carmel. While the training was funded through a research grant, participants were asked to pay a symbolic amount in order to increase their commitment to the process.

*4.3. Sample*

Druze residents of Isfiya and Daliyat el-Carmel were randomly approached in public places and invited to participate in the survey distributed at T1. Two hundred and sixty-five of those approached agreed to participate, a response rate of 84.6%. Respondents were assured anonymity.

Slightly more than half of the respondents (53%) were male. Respondents' average age was 39 years (SD = 13). Seventy-nine percent were married, and all others were unmarried, divorced or widowed. About one-third of the sample (34%) reported that their monthly income was close to the national average (9000 NIS), 41% earned less than the national average and 25% earned above the national average. Most of the respondents (85%) were employed, either working for a firm or business or self-employed (71% and 14%, respectively). Of those who were employed, most (60%) worked locally (i.e., within the two local authorities of Isfiya and Daliyat el-Carmel).

*4.4. Measures*

We collected data on three attitudes towards local sustainability governance related to the Mount Carmel Biosphere Reserve (awareness of the existence of the Mount Carmel Biosphere Reserve, attitudes towards social and economic opportunities in the area of the reserve, resistance to restrictions and environmental enforcement at the reserve), and two intended behaviors (participation and involvement in operating the Mount Carmel Biosphere Reserve, and active resistance to the reserve). Participants indicated the degree to which they agreed with statements regarding the research variables on a scale of 1 (completely disagree) to 6 (completely agree). We controlled for demographic data (gender, age, marital status, income and employment).

During the process of constructing the questionnaire, we asked three external scholars, who specialized in public management, biosphere reserves and economics, to assess the validity of the scales in accordance with our concepts and definitions. This process improved its consistency and reduced the overlap of items. A pilot survey of 50 participants resulted in further minor improvements to the scale. Then, for the whole sample, we tested the conceptions of the scales in line with recommendations in the literature [55] and with accepted tests of construct and predictive validity (exploratory factor analysis, Pearson's coefficient) and reliability tests, i.e., internal consistency and representativeness (Cronbach's α, independent sample t-tests). The scales were verified as reliable and valid measures. The questionnaire can be found in Appendix A.

*4.5. Data Analysis*

Three statistical procedures were conducted to test the research hypotheses. First, to test Hypothesis 1, we examined the linear relationships between the variables using Pearson's coefficient. Next, to test Hypothesis 1 further, we used two hierarchical regressions to examine the contribution of the attitudes to the intended behavioral outcomes, while controlling for demographic data. To test Hypothesis 2, which expected differences in all measures after the training, we used a series of independent sample *t*-tests.

## 5. Findings

*5.1. Relationships between Attitudes Towards and Intended Behaviors Regarding Local Sustainability Governance*

Table 1 presents the mean values, standard deviations, and correlation coefficients for all research variables. The table shows the results for 300 Druze participants (265 residents and 35 Druze entrepreneurs) at T1, before the training.

As is evident from Table 1, the results confirm most of the expected relationships of Hypothesis 1. As predicted, both awareness of the existence of the Mount Carmel Biosphere Reserve and attitudes towards social and economic opportunities in the area of the reserve had moderate positive and significant relationships with participation and involvement in operating the reserve ($r = 0.39$, $p < 0.001$; $r = 0.28$, $p < 0.001$, respectively).

**Table 1.** Correlation matrix for the research variables at T1 (Cronbach's alpha in parentheses).

| Attitudes | Mean | S.D. | 1 | 2 | 3 | 4 | 5 |
|---|---|---|---|---|---|---|---|
| 1. Awareness of the existence of Mount Carmel Biosphere Reserve | 3.18 | 1.53 | (0.86) | | | | |
| 2. Attitude towards social and economic opportunities | 4.15 | 1.01 | 0.48 *** | (0.72) | | | |
| 3. Resistance towards restrictions and environmental enforcement | 3.07 | 1.05 | 0.10 | −0.08 | (0.67) | | |
| **Intended Behaviors** | | | | | | | |
| 4. Participation and involvement in operating the Mount Carmel Biosphere Reserve | 2.05 | 1.27 | 0.39 *** | 0.28 *** | 0.07 | (0.92) | |
| 5. Active resistance towards the Mount Carmel Biosphere Reserve | 1.97 | 1.13 | 0.01 | −0.12 * | 0.43 *** | 0.17 ** | (0.91) |

N = 300; * $p < 0.05$, ** $p < 0.01$, *** $p < 0.001$.

Thus, the more participants were aware of the Mount Carmel Biosphere Reserve and the more positive their attitudes towards social and economic opportunities in the area, the greater their reported participation and involvement in operating the reserve. As expected, both attitudes towards social and economic opportunities in the area of the reserve and resistance to restrictions and environmental enforcement at the reserve had significant relationships with active resistance to the reserve ($r = -0.12$, $p < 0.05$; $r = 0.43$, $p < 0.001$, respectively). Thus, the more positive participants' attitudes towards social and economic opportunities in the area of the reserve were, the less they reported active resistance to the reserve. Similarly, the greater their attitudinal resistance to restrictions and environmental enforcement at the reserve, the greater their intended behavioral resistance. In addition, the attitude measures were correlated with each other, as were the intended behavioral measures. Thus, awareness of the existence of the Mount Carmel Biosphere Reserve was significantly related to attitudes towards social and economic opportunities in the reserve area ($r = 0.48$, $p < 0.001$); and participation and involvement in operating the Mount Carmel Biosphere Reserve was significantly and positively ($r = 0.17$, $p < 0.01$) related to active resistance to the reserve. We will return to these findings in the discussion section.

### 5.2. Predicting Participation and Involvement in Operating the Mount Carmel Biosphere Reserve

To test Hypothesis 1 further, we conducted a hierarchical regression test. Table 2 shows the contributions of the demographic variables and attitudes towards the Mount Carmel Biosphere Reserve to explaining participation and involvement in operating the reserve. Druze residents who were older ($\beta = 0.17$, $p < 0.05$) participated and were involved to a greater extent in operating the reserve. However, attitudes, together with the demographic variables, were better predictors of participation and involvement in operating the reserve than the control variables alone ($F$ for $\Delta R^2 = 12.89$, $p < 0.001$). Awareness of the reserve's existence made the most substantial and the only significant contribution to predicting participation and involvement in operating the reserve ($\beta = 0.34$, $p < 0.001$). In other words, these data provided partial support for Hypothesis 1. The greater the residents' awareness of the existence of the biosphere reserve, the greater their participation and involvement in operating it.

**Table 2.** Hierarchical regression (standardized coefficients $\beta$) for predicting intended behaviors regarding the Mount Carmel Biosphere Reserve (t-test in parentheses).

| | Intended Behaviors Regarding the Mt. Carmel Biosphere Reserve | | | |
| --- | --- | --- | --- | --- |
| | Participation and Involvement in Operating the Mount Carmel Biosphere Reserve | | Active Resistance to the Mount Carmel Biosphere Reserve | |
| | Step 1 | Step 2 | Step 1 | Step 2 |
| Control Variables | | | | |
| Gender | 0.03 (0.33) | 0.03 (0.41) | 0.08 (1.03) | 0.05 (0.72) |
| Age | 0.17 * (2.36) | 0.12 (1.78) | 0.18 * (2.41) | 0.11 (1.66) |
| Education | −0.06 (−0.89) | −0.07 (−1.05) | 0.06 (0.79) | 0.08 (1.18) |
| Salary | 0.05 (0.60) | 0.05 (0.68) | −0.07 (−0.88) | −0.06 (−0.85) |
| Attitudes towards the Mount Carmel Biosphere Reserve | | | | |
| Awareness of the existence of the Mount Carmel Biosphere Reserve | | 0.34 ** (4.51) | | 0.06 (0.77) |
| Attitude towards social and economic opportunities | | 0.11 (1.46) | | −0.16 * (−2.21) |
| Resistance towards restrictions and environmental enforcement | | −0.01 (−0.07) | | 0.39 ** (5.96) |
| $R^2$ | | 0.21 | | 0.23 |
| Adjusted $R^2$ | | 0.18 | | 0.20 |
| $F$ | | 6.98 ** | | 8.11 ** |
| $\Delta R^2$ | | 0.16 | | 0.19 |
| $F$ for $\Delta R^2$ | | 12.89 ** | | 15.65 ** |

Note (N = 285), * $p < 0.05$, ** $p < 0.001$.

### 5.3. Predicting Active Resistance to the Mount Carmel Biosphere Reserve

The results of the hierarchical regression test in Table 2 also show the contributions of the demographic variables and attitudes towards the Mount Carmel Biosphere Reserve to explaining active resistance to the reserve. Druze residents who were older ($\beta = 0.18$, $p < 0.05$) were more likely to actively resist the reserve. Attitudes towards the reserve together with the demographic variables were better predictors of active resistance to the reserve than the control variables alone ($F$ for $\Delta R^2 = 15.65$, $p < 0.001$). Attitudes towards social and economic opportunities in the area of the reserve ($\beta = -0.16$, $p < 0.05$) and resistance to restrictions and environmental enforcement at the reserve ($\beta = -0.39$, $p < 0.001$) made the most substantial and significant contribution to predicting active resistance to the reserve. Once again, the results provide partial support for Hypothesis 1.

*5.4. The Effect of Green Business-Guidance Training on Attitudes Towards and Intended Behaviors of Druze Entrepreneurs Regarding Local Sustainability Governance*

To examine Hypothesis 2, we used three sets of tests. First, to ensure that our experimental and Control groups were comparable, we tested for differences between the general Druze population and the Druze entrepreneurs in the research variables at T1, before the training. For this purpose, we conducted five independent sample t-tests on the research variables (these were previously tested and found to be normally distributed). As expected, the analyses revealed no significant differences between the general Druze population and the Druze entrepreneurs at T1.

Second, to estimate the effects of the training we employed five additional independent sample t-tests. These tests compared the levels of the three attitudes and the two intended behaviors of the entrepreneurs following the training with those of the general population. The means, standard deviations and differences are shown in Table 3.

Looking at attitudes towards the Mount Carmel Biosphere Reserve, significant changes were found for awareness of the existence of the reserve ($t = -1.73$, $p < 0.10$) and attitudes towards social and economic opportunities in the area of the reserve ($t = 2.13$, $p < 0.05$). The average awareness score for the general population at T1 stood at 3.11 ($sd = 1.53$). By the end of the training, the score for the entrepreneurs was 3.68 ($sd = 1.42$). Likewise, the average score for attitudes towards social and economic opportunities in the reserve area rose from 4.12 ($sd = 1.04$) for the general population at T1 to 4.58 ($sd = 0.91$) for the entrepreneurs at T2. A further look at the intended behavior data reveals that active resistance to the reserve fell from a mean of 1.97 ($sd = 1.17$) for the general population at T1 to 1.45 ($sd = 0.46$) for the entrepreneurs after the training ($t = 4.05$, $p < 0.01$).

**Table 3.** Means, standard deviations and differences in attitudes towards and intended behaviors towards regarding the Mount Carmel Biosphere Reserve before and after the training, as calculated by independent sample T-Tests.

| | The General Population at T1 | | Entrepreneurs at T2 | | Difference |
|---|---|---|---|---|---|
| **Attitudes** | **Mean** | **S.D.** | **Mean** | **S.D.** | **T** |
| 1. Awareness of the existence of Mount Carmel Biosphere Reserve | 3.11 | 1.53 | 3.68 | 1.42 | −1.73 * |
| 2. Attitude towards social and economic opportunities | 4.12 | 1.04 | 4.58 | 0.91 | −2.13 ** |
| 3. Resistance to restrictions and environmental enforcement | 3.06 | 1.03 | 2.97 | 1.01 | 0.38 |
| **Intended Behaviors** | | | | | |
| 4. Participation and involvement in operating the Mount Carmel Biosphere Reserve | 2.03 | 1.29 | 2.17 | 1.32 | −0.441 |
| 5. Active resistance to the Mount Carmel Biosphere Reserve | 1.97 | 1.17 | 1.45 | 0.46 | 4.05 *** |

N = 285; * $p < 0.10$, ** $p < 0.05$, *** $p < 0.01$.

## 6. Discussion and Conclusions

This article sought to analyze the effect of biosphere reserve management and green business-guidance training on entrepreneurs' attitudes towards local sustainability governance and their related intended behaviors. A large body of interdisciplinary research in the fields of ecosystem management, sustainability, governance, economic development and entrepreneurship has explored the tension between development and preservation in general, and in relation to its practical manifestation in biosphere reserves. Nevertheless, most of this research has tended to focus on conceptualizing the role of these reserves using case studies. In practice, some biosphere reserves are still battlefields of conflicting interests, policy trends and local stakeholders. As a result, biosphere reserve managements

are struggling to create and sustain the networks and collaborations that promote fruitful ecosystems. This challenge puts biosphere reserves and their delicate goal of fostering conservation and social sustainability along with economic development at risk.

To help fill the theoretical and practical gaps in the literature, and to provide a modest response to the calls of Heinen [12], Heinen and Low [13] and Low and Heinen [14] to promote a practical, flexible approach to conservation programs, we conducted an analysis of empirical data obtained from action research and a before-and-after field experiment to study the relationship between training and Druze entrepreneurs' attitudes towards and intended behaviors regarding sustainability governance. Our study makes three main contributions to the literature. First, we contribute to the longstanding debate about how the interface of urban sociology, economic policy and the regulation of space influences sustainability development and spatial inequality (e.g., [16,17,56]). Second, we add to the literature information about possible conflicts and tensions between conservation and representation, and between the professional management of biosphere reserves and local interests. While many studies in the field of sustainability seek to examine lifestyle changes as a key to achieving sustainability in the environmental aspect, the present study seeks to foster local sustainability through entrepreneurship and strengthening a local economy. We explore whether non-governmental bodies and representatives should have a role in biodiversity conservation decision-making (e.g., [32,33,35]) and how biosphere reserve management can foster sustainability governance (e.g., [9,12,23,25,31]). Third, we add to the literature on the theory of planned behavior (e.g., [35]) and the literature on entrepreneurship training (e.g., [42,49,50]). In other words, we explore the realization of the economic potential inherent in the biosphere reserve that can contribute to strengthening local sustainability in both environmental and community aspects.

Our findings establish two core ideas in the realms of sustainability governance and biosphere reserve management. First, when it comes to sustainability governance, residents' perceptions and their intended behaviors are linked. The more residents of the biosphere reserve are aware of and have positive attitudes towards its social and economic opportunities, the more they participate in operating the biosphere reserve. The opposite is also true. The less aware they are of the reserve and the stronger their rejection of its social and economic opportunities, the less actively they will participate in operating the biosphere reserve. Equally, the more positive the residents' attitudes towards the reserve's social and economic opportunities, and the lower their resistance to sustainability policies, the less they will actively resist the biosphere reserve. Here, again, the opposite is also true. The more they reject the reserve's social and economic opportunities and resist sustainability policies, the greater their active resistance to the presence of the biosphere reserve.

Second, green business-guidance training can highlight the benefits of biosphere reserves for the local community, improving local entrepreneurs' attitudes towards and intended behaviors regarding the concepts of biosphere reserves and sustainability governance. After the training, entrepreneurs became more aware of these concepts and adopted a more positive tone towards the social and economic opportunities related to green businesses. Accordingly, after the training entrepreneurs reduced their active resistance to the presence of the biosphere reserve.

All in all, there is great value in encouraging members of the local population, especially minorities, to increase their involvement and participation in sustainability governance and biosphere reserve management—whether through publishing transparent information, enabling active involvement or supplying education or providing training [25]. Therefore, rather than making top-down declarations about the creation of a new biosphere reserve, and the regulations and requirements that are relevant for anyone visiting, living, working, doing business or even simply passing through the area, the managers of such reserves should combine top-down and bottom-up initiatives to achieve the goals of sustainability governance. In order to improve local stakeholders' attitudes, promote cooperation and prevent active resistance, biosphere reserve managements should initiate collaborations and work with local groups to design, plan and produce policies, regulations and projects that work for both the local community and the biosphere reserve.

Another important implication of our findings is that the values, concepts and practices of the conservation of natural resources are not in and of themselves sufficient for promoting sustainability, at least among local minority entrepreneurs, in whose eyes the creation of the biosphere reserve may hinder economic growth and prosperity. In accordance with scholars who have argued that the residents of biosphere reserves may be more supportive of conservation if they believe they will benefit from it (e.g., [43]), we found that entrepreneurs' attitudes towards and intended behaviors regarded sustainability are strongly affected by their pocketbooks. To win over local business owners and entrepreneurs, conservation values must be linked with economic development and growth. The good news is that we found empirical evidence for practical ways to reduce spatial inequality and the tension between economic development and the preservation of nature. This argument also contributes to the debate about private versus public sector entrepreneurship (e.g., [57]). While some scholars hold that entrepreneurs tend to be interested in personal gain and other scholars claim they want to solve social problems, we emphasize the blurred boundaries between private and public interests, costs and benefits. Thus, we argue that increased awareness and understanding may overcome the problems affecting residents' support for conservation and biosphere reserves (e.g., [2,21]). Specifically, green business-guidance training seems a promising governance tool aimed at fostering sustainability in biosphere reserves.

Notwithstanding the above contributions, our study has some limitations inherent to our methodology. By their nature, our action research and field experiment included a limited number of entrepreneurs and trainees in specific geographic, political and cultural contexts. Therefore, generalizing from this study to other cases should be done cautiously. However, we maintain that when carefully planned and employed by biosphere reserve managements, green business-guidance training may be a practical tool. Likewise, our measures are based on self-reflection, which is inherently subject to bias. Finally, future studies should seek ways to broaden the scope, both theoretically and empirically, of sustainability governance policies, entrepreneurship and training. For instance, future studies may investigate how education programs and training affect the general population and other stakeholders. In addition, future studies may explore how local entrepreneurs translate their attitudes and opinions into commercial initiatives such as green businesses.

**Author Contributions:** I.B. designed, the conceptualization, methodology and research, proposed the model, analyzed the data; and wrote the paper. He also obtained the funding for it. D.G. wrote the paper. I.I. designed the conceptualization, methodology and research, conducted the study, and reviewed and edited the paper. He also helped obtain the funding. T.E. designed the conceptualization, methodology and research, and reviewed and edited the paper. She also helped obtain the funding. N.C. designed the conceptualization, methodology and research, proposed the model, analyzed the data, and was the project administrator. All authors read and approved the final manuscript.

**Funding:** This research was funded by BDU—Deutsche Bundesstiftung Umwelt—and the Israel Ministry of Environmental Protection.

**Acknowledgments:** The authors would like to thank BDU—Deutsche Bundesstiftung Umwelt—and the Israel Ministry of Environmental Protection for funding this study.

**Conflicts of Interest:** The authors declare that they have no conflict of interest. The funders had no role in the design of the study, in the collection, analyses, or interpretation of the data, in the writing of the manuscript, or in the decision to publish the results.

## Appendix A. The Measurement Tool

*Appendix A.1. Awareness of the Existence of the Mount Carmel Biosphere Reserve*

1. I am aware that there is a biosphere reserve in the Carmel
2. I know the borders of the Mount Carmel Biosphere Reserve
3. I know the people and institutions engaged in conserving the environment and am aware of their activities in the Mount Carmel Biosphere Reserve

*Appendix A.2. Attitude Towards Social and Economic Opportunities in the Area of the Mount Carmel Biosphere Reserve*

1. The Mount Carmel Biosphere Reserve creates unique opportunities for economic development for local residents
2. Cooperation between me and the parties responsible for the natural environment in the Mount Carmel Biosphere Reserve can create more profits for me and for the rest of the residents in the area
3. Development of nature and the environment in the Mount Carmel Biosphere Reserve can contribute significantly to the economic development of businesses in the area
4. The activities of environmental organizations in the Mount Carmel Biosphere Reserve are positive and contribute to the development of nature and the environment in the region (r)
5. The natural environment in the Mount Carmel Biosphere Reserve should serve the needs of the community and businesses

*Appendix A.3. Resistance to the Restrictions and Enforcement of The Environmental Regulations in the Mount Carmel Biosphere Reserve*

1. I am opposed to a situation where, because of environmental aspects, I will lose money
2. Conservation of nature and the environment in the Mount Carmel Biosphere Reserve will cause economic damage to the residents of the area
3. People have the right to change nature in accordance with their needs
4. The environmental crisis is exaggerated out of proportion and therefore there is no need to inspect and restrict the business activity of local residents in the Mount Carmel Biosphere Reserve
5. The representatives of environmental organizations in the Mount Carmel Biosphere Reserve should not be allowed to impose fines on residents and business owners in Druze villages

*Appendix A.4. Participation and Involvement in Operating the Mount Carmel Biosphere Reserve*

1. I am in constant contact with representatives of environmental organizations in the Mount Carmel Biosphere Reserve
2. I feel that I am a partner with and affect the management of the Mount Carmel Biosphere Reserve
3. I take part in decision-making about environmental aspects in the Mount Carmel Biosphere Reserve

*Appendix A.5. Active resistance to the Mount Carmel Biosphere Reserve*

1. I am opposed to the existence of the Mount Carmel Biosphere Reserve
2. I work to overturn the restrictions and compromises reached with representatives of environmental organizations in the Mount Carmel Biosphere Reserve
3. My friends and I are working to abolish the Mount Carmel Biosphere Reserve
4. My friends and I are working to dissolve or remove the environmental organizations in the Mount Carmel Biosphere Reserve

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
