# Peer review of "The Impact of Training on Druze Entrepreneurs’ Attitudes Towards and Intended Behaviors Regarding Local Sustainability Governance: A Field Experiment at the Mount Carmel Biosphere Reserve"

_sustainability, doi:10.3390/su12114584_

Round 1
Reviewer 1 Report
This paper is quite sound as is, although there are other aspects of theory that could be addressed. For example, Heinen 1994, Int'l J. Just. Devel. and World Ecology 1:22-33 and related works/. The moths and analysis are quite strong and conclusions are justified. Run through a spell check and English edit.
Author Response
Re: R1: Sustainability-780389:
'The Impact of Training on Druze Entrepreneurs' Attitudes Toward and Intended Behaviors Regarding Local Sustainability Governance: A Field Experiment at the Mount Carmel Biosphere Reserve'
Attached please find our revised article "'The Impact of Training on Druze Entrepreneurs' Attitudes Toward and Intended Behaviors Regarding Local Sustainability Governance: A Field Experiment at the Mount Carmel Biosphere Reserve.” We would like to thank you for the opportunity to improve our article. We would also like to thank the referees for their useful and detailed comments, which we believe made a real contribution to significantly improving the paper and its content. We did our best to satisfy the reviewers and relied on the literature to do so. All in all, we hope that the paper is appropriate for publication in Sustainability. However, should anything be needed, please do not hesitate to contact us. We addressed all of the concerns and questions raised by the reviewers as follows:
Reviewer 1:
This paper is quite sound as is, although there are other aspects of theory that could be addressed. For example, Heinen 1994, Int'l J. Just. Devel. and World Ecology 1:22-33 and related works. The moths and analysis are quite strong and conclusions are justified. Run through a spell check and English edit.
We thank the reviewer for his/her support for this paper. We are also grateful for directing us to Heinen, 1994 and related works such as Heinen and Low (1992) and Low and Heinen (1993). In the revised version we used these references to better explain our theoretical framework (p. 2: 64-72; p.3: 79), strengthen the research model and hypotheses (p. 6: 282-291; p. 7: 328-340) and deepen the discussion and conclusion in response to the existing literature (p. 12: 536-549; p. 13: 570-577). We also sent the paper to a professional English-language editor.
Reviewer 2 Report
This is a very interesting and well-presented paper. i have a few comments only below, but the use of protection and conservation are used interchangeably sometimes - for BRs conservation is the correct term to use throughout. this has interest and use to the BR community - on publication I would urge yo to make the MAB Secretariat in Paris aware of it.
L 30 they are nominated by national govts but managed by a varying array of systems.
L35 agree
L102/3 protection is not conservation – this needs clarification
L108 et seq – I would say BRs are sites for sustainable development with conservation cores integrated into their management.
L112 BRs were established in 1971 not the ‘60s
L128 again its conservation not protection – suggest the ms is read carefully in terms of using these terms.
L165 suggest local community rather than locals’(sic)
L228 – the map is very clear but also interesting in that it should not technically be part of the world network as there are 2 core areas that seem to be outside the BR boundaries – is this correct?
Author Response
Re: R1: Sustainability-780389:
'The Impact of Training on Druze Entrepreneurs' Attitudes Toward and Intended Behaviors Regarding Local Sustainability Governance: A Field Experiment at the Mount Carmel Biosphere Reserve'
Attached please find our revised article "'The Impact of Training on Druze Entrepreneurs' Attitudes Toward and Intended Behaviors Regarding Local Sustainability Governance: A Field Experiment at the Mount Carmel Biosphere Reserve.” We would like to thank you for the opportunity to improve our article. We would also like to thank the referees for their useful and detailed comments, which we believe made a real contribution to significantly improving the paper and its content. We did our best to satisfy the reviewers and relied on the literature to do so. All in all, we hope that the paper is appropriate for publication in Sustainability. However, should anything be needed, please do not hesitate to contact us. We addressed all of the concerns and questions raised by the reviewers as follows:
Reviewer 2:
This is a very interesting and well-presented paper.
We thank the reviewer for his/her support for this paper.
I have a few comments only below, but the use of protection and conservation are used interchangeably sometimes - for BRs conservation is the correct term to use throughout.
Reviewer 2 is right that protection is unsuitable for the context of biosphere reserves. Accordingly, we carefully re-read the manuscript and replaced all of the inappropriate uses of 'protection' with 'conservation'.
This has interest and use to the BR community - on publication I would urge you to make the MAB Secretariat in Paris aware of it.
We thank reviewer 2 for this encouraging comment. We will definitely follow up on this recommendation after publication.
L 30 they are nominated by national govts but managed by a varying array of systems.
Corrected (p. 1:32).
L35 agree
L102/3 protection is not conservation – this needs clarification
Following the first comment, we fixed the inappropriate use of protection. In this case we deleted "(or “protection”)" (p. 3:107).
L108 et seq – I would say BRs are sites for sustainable development with conservation cores integrated into their management.
Corrected (p. 3:113-114).
L112 BRs were established in 1971 not the ‘60s
Corrected (p. 3:117).
L128 again its conservation not protection – suggest the ms is read carefully in terms of using these terms.
Following the first comment, we carefully re-read the manuscript and replaced all of the inappropriate uses of 'protection' with 'conservation'.
L165 suggest local community rather than locals’(sic)
Corrected. We replaced locals with local community (p. 4: 185).
L228 – the map is very clear but also interesting in that it should not technically be part of the world network as there are 2 core areas that seem to be outside the BR boundaries – is this correct?
We double checked the accuracy of our map. Following reviewer 2’s comment, we replaced it with a new map that better depicts the external and internal boundaries of the BR and the zones (p. 5).
Reviewer 3 Report
This could potentially be an interesting article. However, I feel that that the authors need to explain the rationale for the research better, and need to provide more information on the training provided.
The authors hypothesize, and claim to have found evidence for, increased positive attitudes and behaviors towards sustainability after providing Druze entrepreneurs with a training on green entrepreneurship. The basis for assuming this link is not well explained. The authors need to explain how they view and define sustainability governance, and why this is linked to entrepreneurship. The authors do explain that once people are better able to (identify) economic benefits from conservation they will be more supportive of conservation, but is that really the same as supporting sustainability governance? Do they link sustainability governance in any way to the lack of involvement of local residents in the management of the biosphere reserve?
The authors also claim that the training results in behavioral changes, but from the questionnaire provided in the annex, it is not entirely clear how this is measured. The questions are quite vague - only one question related to resistance refers to behavior, and only quite generally and vaguely... The way participation is measured also seems quite problematic, especially given the discussion in the introduction of the paper lamenting the lack of any local involvement in the management of the biosphere reserve. Question 3 asks about involvement in decision-making, while in the introduction the authors state that residents do not have a say in the management of the biosphere reserve - so what are the environmental aspects referred to in that question?
Detailed comments:
- Lines 147-155: the critique by Backstrand needs to be explained more clearly - e.g. does it increase democratic deficits, and how?
- Line 156: I would say Contrary to, rather than in addition...
- Lines 158-160: the impairment model needs to be briefly explained
- Paragraph 3.1. really needs to start with a discussion of how the authors define sustainability governance in general and in relation to the specific site, and how this is related to entrepreneurship. There is some explanation regarding economic benefits, but this is inadequate.
- Lines 319-325: this really needs to be explained in more detail
- Lines 525-529: these are some bold claims, and I wonder whether the authors actually can substantiate these. Though I am very much in favor of combining conservation and development, I don't think the study offers any proof that this is option is possible or the best possible option, as the behavior measures (self-reported) are quite limited, and no study was conducted on the actual impact of entrepreneurial activities on the environment, and the statement about fostering sustainability governance really hinges on how one defines that...
Author Response
May 5, 2020
Dear Prof. Dr. Marc A. Rosen
Editor-in-Chief,
Sustainability
Re: R1: Sustainability-780389:
'The Impact of Training on Druze Entrepreneurs' Attitudes Toward and Intended Behaviors Regarding Local Sustainability Governance: A Field Experiment at the Mount Carmel Biosphere Reserve'
Attached please find our revised article "'The Impact of Training on Druze Entrepreneurs' Attitudes Toward and Intended Behaviors Regarding Local Sustainability Governance: A Field Experiment at the Mount Carmel Biosphere Reserve.” We would like to thank you for the opportunity to improve our article. We would also like to thank the referees for their useful and detailed comments, which we believe made a real contribution to significantly improving the paper and its content. We did our best to satisfy the reviewers and relied on the literature to do so. All in all, we hope that the paper is appropriate for publication in Sustainability. However, should anything be needed, please do not hesitate to contact us. We addressed all of the concerns and questions raised by the reviewers as follows:
Reviewer 3:
This could potentially be an interesting article. However, I feel that that the authors need to explain the rationale for the research better, and need to provide more information on the training provided.
We are grateful to reviewer 3 for his/her insightful comment. Based on it, we made several significant improvements that we feel improved the paper a great deal. First, we defined sustainability governance and improved the explanation of this concept in our study’s context (p. 3: 123-132; p. 4: 144-149). Second, we added a whole new paragraph under a new section 3.1 called 'Rationale for the Study' (p. 6: 244-286). This section details the rationale and assumptions of our study and the concept of local entrepreneurs. It describes our assumptions about the relationship between the entrepreneurs, the training, and the expected change in attitudes and intended behaviors. In addition, it lays the groundwork for the path of our argument that results in our two hypotheses (now 3.2 and 3.3).
The authors hypothesize, and claim to have found evidence for, increased positive attitudes and behaviors towards sustainability after providing Druze entrepreneurs with a training on green entrepreneurship. The basis for assuming this link is not well explained. The authors need to explain how they view and define sustainability governance, and why this is linked to entrepreneurship.
Following the reviewer’s suggestions, we defined sustainability governance and improved the explanation of this concept in our study’s context. Second, we also muted our claim about behavior to intended behavior. We now provide a more modest and cautious description of the relationships. Third, we improved the basis for assuming this link in several sections, particularly in sections 3.1, 3.2 and 3.3 (p. 3: 123-131; p. 4: 144-149; p. 6: 277-291; p. 8: 321-342).
The authors do explain that once people are better able to (identify) economic benefits from conservation they will be more supportive of conservation, but is that really the same as supporting sustainability governance? Do they link sustainability governance in any way to the lack of involvement of local residents in the management of the biosphere reserve?
Thanks for this comment. Accordingly, we described the rationale for the study and referred to supporting sustainability governance and to the level of participation. We also improved the argumentation as described earlier. In addition, we added more explanations in the method section to clarify the variables that were measured and the role of the trainers. In addition, we also stressed these points in the description of the training (p. 9: 379-384, 392-417).
The authors also claim that the training results in behavioral changes, but from the questionnaire provided in the annex, it is not entirely clear how this is measured. The questions are quite vague - only one question related to resistance refers to behavior, and only quite generally and vaguely... The way participation is measured also seems quite problematic, especially given the discussion in the introduction of the paper lamenting the lack of any local involvement in the management of the biosphere reserve. Question 3 asks about involvement in decision-making, while in the introduction the authors state that residents do not have a say in the management of the biosphere reserve - so what are the environmental aspects referred to in that question?
Following this comment, we muted our claim about behavior to intended behavior. We now provide a more modest and cautious description of the relationships between attitudes towards the BR and intended behavior regarding it and sustainability governance. Regarding participation and involvement in decision-making, we provided a detailed rationale for the study in section 3.1 (p. 6: 244-268). According to this rationale, perceptions and attitudes (both positive and negative) towards sustainability governance probably exist, whether the management involves the public in its decisions or not. Generally speaking, the broader public is not involved in the management of the biosphere reserve to any great degree. However, according to the rationale, taking part in the training, which is provided on behalf of the formal conservation authorities, should promote positive attitudes towards the BR and sustainability governance. In addition, taking part in the training involves more professional meetings, discussions and brainstorming with the authorities' decision makers and policy makers, which may increase the perception of being involved in decision-making. We stressed these points in the description of the training (p. 9: 379-384, 392-417).
Detailed comments:
Lines 147-155: the critique by Backstrand needs to be explained more clearly - e.g. does it increase democratic deficits, and how?
Corrected. We rephrased this paragraph. The explanation of the tension between participation and democratic deficits and between professionalism and representation is now clearer (p. 4: 160-167).
Line 156: I would say Contrary to, rather than in addition...
Corrected. We replaced 'In addition' with 'Contrary to the widespread expectations' (p. 4: 172).
Lines 158-160: the impairment model needs to be briefly explained
Corrected. We expanded the explanation of the impairment model (p. 4: 175-179).
Paragraph 3.1. really needs to start with a discussion of how the authors define sustainability governance in general and in relation to the specific site, and how this is related to entrepreneurship. There is some explanation regarding economic benefits, but this is inadequate.
As we detailed above, we defined sustainability governance and improved the explanation of this concept in our study’s context (p. 3: 123-129; p. 4: 144-149). We added a whole new paragraph under a new section 3.1 called 'Rationale for the Study' (p. 6: 233-268). Inter alia, this section details the rationale and assumptions of our study and the concept of local entrepreneurs. In addition, it familiarizes the reader with the path of our argument that is followed in the next section by our two hypotheses (now 3.2 and 3.3).
Lines 319-325: this really needs to be explained in more detail
We expanded our explanation of this point: (p. 8: 328-339): “Local entrepreneurs are motivated to and skilled at accomplishing societal and commercial goals. They know how to exploit opportunities and use innovative ideas and strategies to influence their surroundings and improve their self-interests [39]. Accordingly, we expect that if the conservation authorities provide training to local entrepreneurs, including meetings and sessions with the authorities' decision makers and policy makers, they will improve the local entrepreneurs' attitudes toward and intended behaviors regarding sustainability governance. Furthermore, we claim that since a change in moral belief alone is insufficient for solving environment problems and changing behaviors, individuals and communities should be provided with other rewards and alternatives [12]. One example is green business training guidance designed to provide rewards for the local community. As studies have demonstrated, the greater the rewards, the more likely the success in changing attitudes toward and intended behavior regarding sustainability [13].”
Lines 525-529: these are some bold claims, and I wonder whether the authors actually can substantiate these. Though I am very much in favor of combining conservation and development, I don't think the study offers any proof that this is option is possible or the best possible option, as the behavior measures (self-reported) are quite limited, and no study was conducted on the actual impact of entrepreneurial activities on the environment, and the statement about fostering sustainability governance really hinges on how one defines that...
Reviewer 3 is right. This claim was exaggerated. Thus, we muted our statement and rephrased our theoretical contribution to be more modest (p.14:536-549)
Dr. Dan Gottlieb
Teaching Fellow, School of Political Sciences, University of Haifa, Israel
Environmental education and sustainability coordinator at Haifa Bay Municipal Association for Environmental Protection
Dr. Noam Cohen
Head, Strategic Planning and Research Department, Emek Yizrael Regional Council
Round 2
Reviewer 3 Report
The author(s) have carefully considered my earlier comments. In the revised version they explain the link they perceive between sustainability governance and entrepreneurial possibilities and training - I am not sure I agree with the way the link is made, but that is another issue. The author(s) have also adapted the earlier claim that the training influenced behavior, to the - substantiated - claim that they influenced attitudes and intended behavior. I am also happy to see that the authors also address the issue of the need to foster local participation, on the basis of the outcomes of their research (though this seems to be a bit in contradiction with what the author(s) state in lines 251-253 in the methods section).
A few minor details:
Line 133: I'm not sure what you mean by the long arm of the government - do you mean that ultimately, (national) governments are responsible for BR management and regulation?
Lind 543-544: please check the formulation of the sentence staring with 'Second, we add...' It seems some words are missing from this sentence
Author Response
Reviewer 3:
I am also happy to see that the authors also address the issue of the need to foster local participation, on the basis of the outcomes of their research (though this seems to be a bit in contradiction with what the author(s) state in lines 251-253 in the methods section).
The reviewer is right. In order to keep the argument coherent, we rephrased lines 251-253 (p. 6): "In addition, this effect is more likely to be positive if the management involves the public in its decisions."
A few minor details:
Line 133: I'm not sure what you mean by the long arm of the government - do you mean that ultimately, (national) governments are responsible for BR management and regulation?
Corrected: We rephrased this sentence to make it clearer: "…all biosphere reserves are designated by the national government and are expected to fulfill…"
Line 543-544: please check the formulation of the sentence staring with 'Second, we add...' It seems some words are missing from this sentence
Corrected: We omitted the word 'the': "Second, we add to the literature about possible conflicts and tensions between conservation and representation, and between the professional management of biosphere reserves and local interests."
